# Structural change as a key component for agricultural non-CO$_2$ mitigation efforts

Stefan Frank[1], Robert Beach[1,2], Petr Havlík[1], Hugo Valin[1], Mario Herrero[3], Aline Mosnier[1], Tomoko Hasegawa[1,4], Jared Creason[5], Shaun Ragnauth[5] & Michael Obersteiner[1]

Agriculture is the single largest source of anthropogenic non-carbon dioxide (non-CO$_2$) emissions. Reaching the climate target of the Paris Agreement will require significant emission reductions across sectors by 2030 and continued efforts thereafter. Here we show that the economic potential of non-CO$_2$ emissions reductions from agriculture is up to four times as high as previously estimated. In fact, we find that agriculture could achieve already at a carbon price of 25 \$/tCO$_2$eq non-CO$_2$ reductions of around 1 GtCO$_2$eq/year by 2030 mainly through the adoption of technical and structural mitigation options. At 100 \$/tCO$_2$eq agriculture could even provide non-CO$_2$ reductions of 2.6 GtCO$_2$eq/year in 2050 including demand side efforts. Immediate action to favor the widespread adoption of technical options in developed countries together with productivity increases through structural changes in developing countries is needed to move agriculture on track with a 2 °C climate stabilization pathway.

[1] International Institute for Applied Systems Analysis, Schlossplatz 1, Laxenburg 2361, Austria. [2] RTI International, Research Triangle Park, 3040 East Cornwallis Road, Durham 27709-2194 NC, USA. [3] Commonwealth Scientific and Industrial Research Organization, 306 Carmodi Road, St Lucia QLD 4067, Australia. [4] Center for Social & Environmental Systems Research, National Institute for Environmental Studies, 16-2 Onogawa, Tsukuba-City 305-8506 Ibaraki, Japan. [5] Environmental Protection Agency, 1200 Pennsylvania Avenue, N.W., Washington 20460 DC, USA. Correspondence and requests for materials should be addressed to S.F. (email: frank@iiasa.ac.at)

At the Conference of the Parties (COP) 21 in Paris, countries adopted a goal to keep global warming well below 2 °C and possibly below 1.5 °C. To achieve this target, cumulative anthropogenic greenhouse gas (GHG) emissions should not exceed 400–1000 GtCO2eq by the end of the century, which makes a rapid adoption of stringent mitigation policies indispensable[1]. To facilitate the distribution of mitigation efforts across countries and monitor progress toward climate stabilization, countries submitted nationally determined contributions (NDCs), which specify nationally-anticipated GHG mitigation and climate change adaptation policies. Agriculture is one of the largest emission sources, accounting for up to 25% of global GHGs[2] and the majority of emissions in many developing countries. Most countries mentioned agriculture in their NDCs, even though a formal negotiation process on mitigation in agriculture is yet to be specified under the United Nations Framework Convention on Climate Change. Agricultural mitigation options can be grouped into options targeting either the supply or the demand side[3,4]. While demand side oriented options target consumer behavior to reduce consumption of GHG-intensive products and waste[5–7], supply side options attempt to improve GHG efficiency of agricultural production. The latter can be split into technical and structural options[3]. Technical options reduce agricultural emissions using technologies like anaerobic digesters, feed supplements, nitrogen inhibitors, etc.[8], to reduce emissions, whereas structural options usually refer to more fundamental adjustments within the agricultural sector such as transition towards high intensity management systems or relocation of production across regions through international trade[9]. Several recent studies have quantified the mitigation potential of agriculture for a subset of options[3,8–13]. The fifth Assessment Report (AR5) of the Intergovernmental Panel on Climate Change (IPCC)[2] estimates an economic bottom-up supply side mitigation potential of 0.3–0.6 GtCO2eq/year (at 100 $/tCO2eq) for agricultural CH4 and N2O emissions in 2030 using a set of technical options based on Smith et al.[4], which is 5%–10% reduction of current emissions.

Wollenberg et al.[14] stress the need to bridge the scientific gap between regional mitigation potentials from bottom-up studies with global mitigation requirements for climate stabilization and propose an aspirational mitigation target of 1 GtCO2eq/year for agriculture by 2030 to be consistent with 2 °C climate stabilization estimated by Integrated Assessment Models (IAMs). However, the existing literature on bottom-up assessment of the agricultural mitigation potential neither considers structural measures on the supply side nor market feedbacks on the demand side. Hence, existing estimates may significantly underestimate the potential contribution of agriculture to global mitigation due to the narrow focus on a subset of options or may overestimate the potential of certain options due to the absence of market feedbacks. Understanding the relative costs of GHG reductions across sectors is vital for achieving mitigation goals cost-effectively. While agricultural CO2 emissions from land use change and soil carbon may be mitigated at relatively low costs[4,15,16], residual agricultural non-CO2 emissions will play a crucial role and determine amongst other factors such as speed of decarbonization and efforts in other sectors, the absolute level of negative emissions required to achieve ambitious climate stabilization targets.

Despite the importance of non-CO2 emissions for achieving ambitious climate stabilization targets, an integrated assessment of the main agricultural mitigation mechanisms at the global scale that incorporates specific mitigation options defined in the bottom-up studies is currently missing. In this study, we quantify the global agricultural CH4 and N2O mitigation potential using a comprehensive set of technical and structural mitigation options on the supply side, and market feedbacks through consumption and international trade responses to price changes. Applying the GLObal BIOsphere Management (GLOBIOM) economic partial equilibrium land use model[9], we estimate the integrated marginal abatement cost curve (MACC) for agricultural non-CO2 emissions. We disaggregate the estimated mitigation potential by region and mitigation mechanism to quantify the importance of technical options, structural changes in agriculture, and consumers' response and identify suitable mitigation pathways for agriculture. We find that agriculture could contribute non-CO2 emission savings of around 1 GtCO2eq/year by 2030 already at 25 $/tCO2eq mainly through the deployment of technical and structural mitigation options in the livestock sector. By 2050, joint efforts on the supply and demand side could allow achieving up to 2.6 GtCO2eq/year of non-CO2 mitigation at 100 $/tCO2eq with potential synergies of around 0.7 GtCO2eq/year for land use change CO2 mitigation.

## Results

**Agricultural non-CO2 emissions without mitigation efforts.** Under a baseline scenario (Shared Socio-Economic Pathway 2, SSP2)[17,18] without mitigation efforts, we project global agricultural CH4 and N2O emissions to increase from 4.8 GtCO2eq/year in 2010 to around 6.8 GtCO2eq/year by 2050 (see also Supplementary Figs 1 and 2). In this scenario, world population increases to around 9.2 billion and gross domestic product (GDP) per capita doubles globally by 2050 driving an increase in global mean calorie intake to around 3200 kcal/capita/day in 2050. Agricultural CH4 and N2O baseline emission projections are close to FAOSTAT estimates (around 6.2 GtCO2eq/year in 2050 applying the global warming potentials from IPCC AR4) and Bennetzen et al.[19] (around 7 GtCO2eq/year in 2050). Compared to other IAMs which project agricultural non-CO2 emissions to increase to 7.6–10.5 GtCO2eq/year in SSP2 by 2050[20] our estimate is more conservative. Pressure from sustained population and GDP growth in Asia are one of the main drivers for expansion of emissions intensive agricultural production in that region (+1.1 GtCO2eq/year compared to 2010 levels). Significant agricultural emission increase is also anticipated for Latin America (+0.3 GtCO2eq/year) and Africa (+0.3 GtCO2eq/year), which can be attributed primarily to the expansion of ruminant production. Non-CO2 emissions in developed regions like Europe, North America, and Oceania are projected to grow only moderately until 2050. By 2050, Asia, Latin America, and Africa are responsible for almost 80% of global N2O and CH4 emissions from agriculture in the absence of mitigation policy.

**Non-CO2 mitigation potential in agriculture.** We calculate mitigation potentials in 2030 and 2050 by applying a series of uniform global carbon prices ranging from 10 to 150 $/tCO2eq. Figure 1 presents the estimated non-CO2 mitigation potentials for selected carbon price ranges for a 2 °C scenario and compares them to other studies and an aspirational mitigation target of 1 GtCO2eq/year in 2030[14] and 2 °C mitigation requirements estimated by IAMs in 2050[20]. We find that the aspirational target for 2030 could be achieved at only 25 $/tCO2eq, which would allow transforming agriculture to be consistent with 2 °C climate stabilization in the short run. Our results indicate a mitigation potential for agricultural non-CO2 emissions in 2030 that is up to four times higher (2 GtCO2eq/year for 100 $/tCO2eq) at the same carbon price than estimated by the IPCC AR5[2] and other bottom-up studies[4,8,21] that do not consider an integrated representation of technical options, structural changes, and consumers' response. In total, technical options contribute around 0.8 GtCO2eq/year at 100 $/tCO2eq in 2030. Estimates for technical mitigation options range from 0.06–0.1 GtCO2eq/year for improved fertilizer

management[4,8], 0.2–0.35 for improved rice cultivation[4,8,22], 0.04–0.1 GtCO$_2$eq/year for manure management[4,8], and 0.03–0.1 GtCO$_2$eq/year for livestock feed supplement[21,23]. The lower mitigation potential of 0.5 GtCO$_2$eq/year in Beach et al.[8], can be explained by more conservative assumptions on adoption shares and the limited applicability of mitigation options on a subset of crops. In the absence of an economic model of adoption, they assume equal adoption rates across the technologies within a mitigation bundle, i.e., different anaerobic digesters have to get

adopted with equal shares, whereas GLOBIOM finds greater adoption of options that provide higher levels of mitigation. Similarly, Vermont and De Cara[13] show in a meta-analysis higher non-CO$_2$ reduction shares in equilibrium models compared to bottom-up or supply side focused model driven by more flexibility in resource allocation. Moreover, Beach et al.[8] apply mitigation options only to a subset of crops (61% of non-rice cropland) whereas in this study we allow options also to be applied to other similar crops represented in GLOBIOM (more details are provided in the SI). However, comparing relative emission reductions of technical options to baseline levels, both studies yield consistent results (around 13% reduction at 100 $/tCO$_2$eq in 2030).

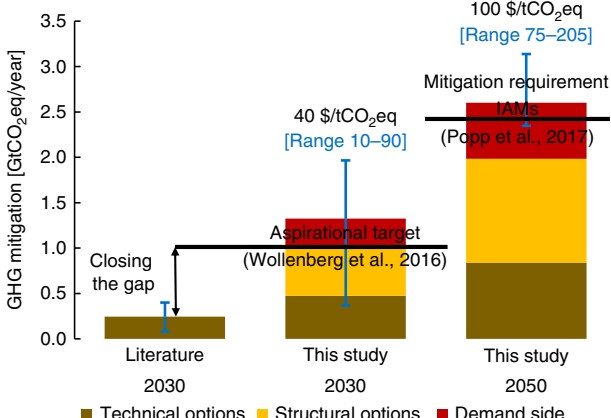

**Fig. 1** Non-CO$_2$ mitigation potential in agriculture. Comparison of agricultural non-CO$_2$ mitigation potentials at 40 and 100 $/tCO$_2$eq with other literature and the aspirational target of 1 GtCO$_2$eq/year by 2030 (in ref. [14]) and mitigation requirements in IAMs by 2050 (in ref. [20]). The average mitigation potential from the literature was calculated across in refs. [4,8,21] at 40 $/tCO$_2$eq. Carbon price ranges and agricultural mitigation requirements by 2050 are consistent with a 2 °C scenario estimated by IAMs (in ref. [20]). Error bars indicate either the minimum and maximum potential across literature studies or the corresponding non-CO$_2$ mitigation potentials in this study given the carbon price range

**Decomposition of agricultural non-CO$_2$ mitigation**. We now want to focus on the 2050 time horizon and elaborate in detail how and at what costs agriculture could contribute to climate stabilization. At the 100 $/tCO$_2$eq carbon price considered by IPCC (2014), we find agricultural non-CO$_2$ emissions could be reduced by up to 2.6 GtCO$_2$eq in 2050 considering both technical and structural supply side mitigation options as well as consumers' response to price changes. This corresponds to non-CO$_2$ emission reductions of almost 40% relative to the baseline. Around 70% of the potential mitigation originates from a reduction of CH$_4$ emissions and 30% from reductions in N$_2$O. Figure 2 presents the agricultural mitigation potential in 2050 by region (2a) and mitigation mechanism (2b) as a function of carbon price. Asia and Latin America offer particularly significant potential for emission reductions, while developed regions like North America, Europe and Oceania can contribute to a much lesser extent due to their lower baseline GHG emission intensity and limited share of global non-CO$_2$ emissions by 2050.

Technical options can provide direct emission reductions of around 0.85 GtCO$_2$eq/year at a carbon price of 100 $/tCO$_2$eq, accounting for around 33% of the total agricultural mitigation globally. However, including indirect effects of these options

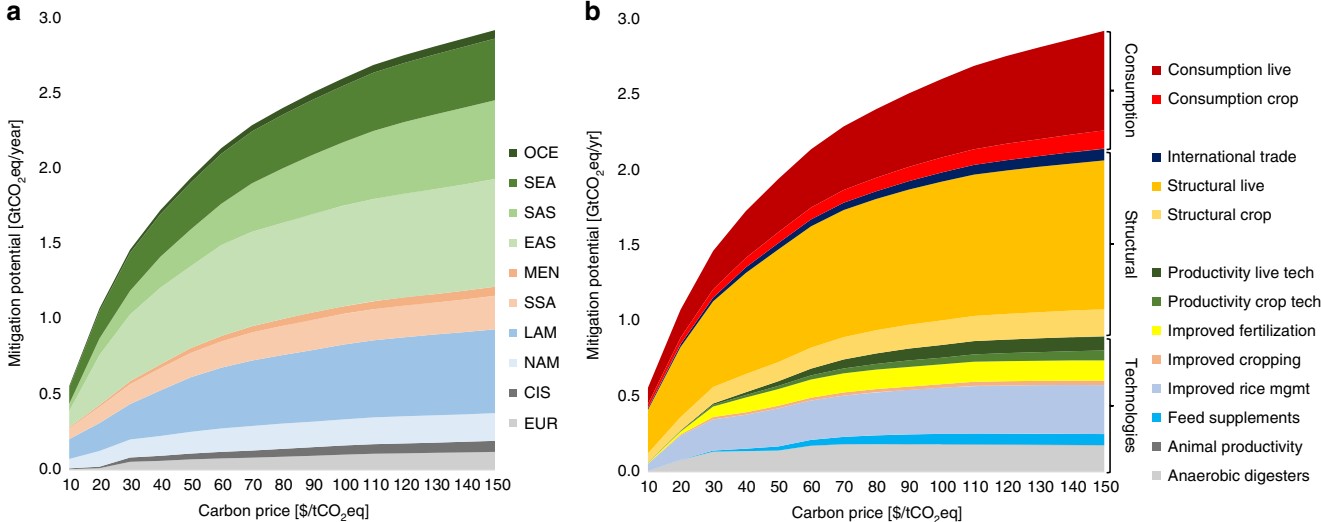

**Fig. 2** Economic supply of non-CO$_2$ mitigation in agriculture. Mitigation potentials are presented by region (**a**) and mitigation option (**b**) in 2050 at increasing global carbon price levels. Acronyms in **a** are: EUR Europe, CIS Commonwealth of Independent States, NAM North America, LAM Latin and Central America, SSA Sub-Saharan Africa, MEN Middle East and North Africa, EAS East Asia, SAS South Asia, SEA Southeast Asia, OCE Oceania. Mitigation options in **b** include consumption changes to price signals; structural options such as livestock and crop system transition, reallocation of production through intra and international trade; and technical options (anaerobic digesters, animal supplements such as antibiotics, bovine somatotropin, propionate precursors, and anti-methanogen vaccination, improved rice management in terms of different combinations of water, residue, and fertilizer management, improved cropping practices such as no tillage and residue incorporation, and improved fertilization practices such as nitrogen inhibitors and optimal fertilizer application)

through related productivity increases adds 0.15 GtCO$_2$eq/year in mitigation and increases the share to 38% of total global mitigation. For example, propionate precursors or anti-methanogen vaccination do not only reduce CH$_4$ emissions from enteric fermentation through improved digestibility but also enhance animal productivity. Hence, more livestock products may be produced with less emissions as the emission intensity (emissions per unit of output produced) decreases. Adoption of these technical options would require investment and operation costs of around 13 bn $/year globally by 2050 (12 bn $ in 2030), the majority arising in emerging and developing regions like Asia (almost half) or Sub-Saharan Africa (10%) while only one quarter of the costs occur in developed countries (Europe, Oceania, and North America). Structural adjustments, i.e., shift from rather GHG inefficient extensive grazing systems toward mixed grass–cereal feeding systems, and could contribute 1.0 GtCO$_2$eq/year and consumer response to prices will add another 0.6 GtCO$_2$eq/year at 100 $/tCO$_2$eq in 2050.

The cost-efficient contribution of the different mitigation options to the total potential varies across regions and at different carbon prices, hence "one size fits all" policies will not enable to realize mitigation potentials cost-efficiently across regions. Figure 3 provides the disaggregated regional mitigation potentials for carbon prices of 40 and 100 $/tCO$_2$eq. Technical mitigation options contribute most significantly (in relative shares) in the developed regions of Europe (80% of total potential, 100 $/tCO$_2$eq) and North America (50% of total potential, 100 $/tCO$_2$eq), mainly through the adoption of highly (cost-) efficient (large-scale) manure management and nitrogen fertilization technologies. In the remaining regions, technical options account for around one quarter of the total mitigation potential. While at lower carbon prices mainly improved rice management options and anaerobic (large-scale) digesters are being adopted, improved fertilization management becomes profitable with rising carbon prices. In Asia, improved rice management such as switching to dryland rice (with residue incorporation) and reduced nitrogen fertilization, offers opportunities to significantly reduce CH$_4$ emissions of up to 0.3 GtCO$_2$eq/year at 100 $/tCO$_2$eq (more than

50% emissions reduction) from flooded rice paddies. Similarly, Hussain et al.[22] highlighted the significant potential for GHG reduction, comparing to traditional rice production systems through improved tillage, irrigation, and fertilization practices of up to 67%.

Structural adjustments such as shifts in production systems or relocation through international trade account for around 39% (1.0 GtCO$_2$eq/year) of the total mitigation potential at 100 $/tCO$_2$eq. Especially in Latin America and East Asia, transition of livestock production systems may significantly reduce non-CO$_2$ emissions. We find mitigation of up to 0.5 GtCO$_2$eq/year in 2050 at 100 $/tCO$_2$eq through the decrease of ruminant production in tropical areas and shift to mixed-cereal feeding systems in temperate areas with higher productivities. In Latin America, this development coincides with a decrease in extensive grassland-based systems. The importance of this transition of ruminant livestock production systems for climate change mitigation is also acknowledged by other studies[3,9,24]. Many of these structural changes are highly cost-efficient. Thus, the mitigation from structural adjustments tends to account for a larger share of total agricultural mitigation potential at lower carbon prices, i.e., structural adjustments provide about half of the total mitigation at 40 $/tCO$_2$eq.

Reduction in consumption levels due to price increases accounts for around 24% (0.6 GtCO$_2$eq/year) of the mitigation potential at 100 $/tCO$_2$eq. Carbon price induced commodity price increases drive consumers to reduce their consumption levels of GHG-intensive products in our modeling framework, mainly in Latin America and South Asia. Global average calorie intake decreases from around 3300 kcal/capita/day to around 3200 kcal/capita/day (−3%) in 2050 while average agricultural commodity prices increase by 18%. However, calculated mitigation potentials take into account a food security constraint, which limits the number of people undernourished in each region. In developed regions such as Europe and North America, consumers are less impacted due to more efficient production systems and therefore less significant price increases, as well as lower demand elasticities related to higher income levels. As the

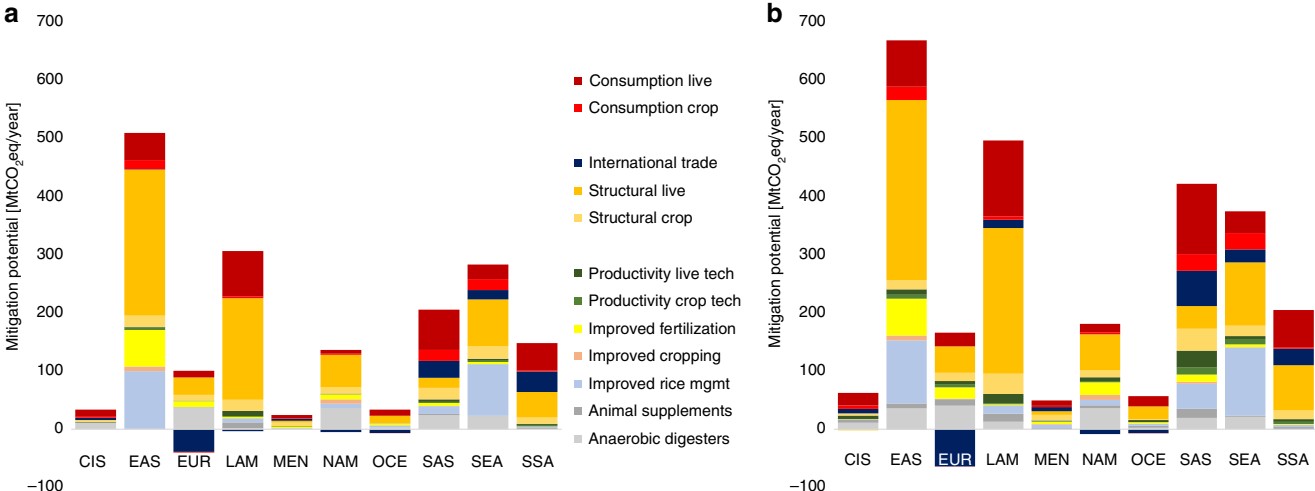

**Fig. 3** Economic supply of regional non-CO$_2$ mitigation in agriculture. Results are displayed for different mitigation options in 2050 for a carbon price of **a** 40 $/tCO$_2$eq and **b** 100 $/tCO$_2$eq. Acronyms are: EUR Europe, CIS Commonwealth of Independent States, NAM North America, LAM Latin and Central America, SSA Sub Saharan Africa, MEN Middle East and North Africa, EAS East Asia, SAS South Asia, SEA Southeast Asia, and OCE Oceania. Mitigation options include consumption changes to price signals; structural options such as livestock and crop system transition, reallocation of production through intra and international trade; and technical options (anaerobic digesters, animal supplements such as antibiotics, bovine somatotropin, propionate precursors, anti-methanogen vaccination, and intensive grazing, improved rice management in terms of different combinations of water, residue, and fertilizer management, improved cropping practices such as no tillage and residue incorporation, and improved fertilization practices such as nitrogen inhibitors and optimal fertilizer application)

mitigation potential from technical and structural options becomes exhausted with increasing carbon price, additional mitigation is mainly achieved through demand side adjustments as consumption decreases with increasing prices. Across the three mitigation wedges, the livestock sector accounts for around 70% (at 100 $/tCO$_2$eq) of the total potential, mostly coming from structural adjustments and reduction in consumption levels. This highlights the importance of livestock for climate change mitigation and its role as a land use driver and source of non-CO$_2$ emissions.

**Potential synergies for CO$_2$ mitigation**. We find that a mitigation policy targeting only non-CO$_2$ emissions from agriculture yields synergies with CO$_2$ mitigation through avoided land use change. Results show additional reduction of 0.7 GtCO$_2$eq/year (100 $/tCO$_2$eq) of CO$_2$ in 2050 due to land sparing through productivity increases, and reduces consumption and production levels. This is in line with other studies who find co-benefits of non-CO$_2$ mitigation through intensification for CO$_2$ emissions from land use change[12,25]. Synergies with CO$_2$ mitigation come on top of the 2.6 GtCO$_2$eq/year reduction in CH$_4$ and N$_2$O and play an important role especially in regions with high land use change emissions such as in Sub-Saharan Africa, Latin America, and Southeast Asia. As GHG-intensive products such as ruminant meat become relatively more expansive with increasing carbon prices especially in regions with low productivity, expansion of extensive pastures decreases significantly in developing regions (Fig. 4). Globally, around 35 million hectare (Mha) of cropland and 225 Mha of pastures are freed up from agricultural use by 2050 at a carbon price of 100 $/tCO$_2$eq on non-CO$_2$ emissions only which could even provide further co-benefits for carbon sequestration in soil and biomass through revegetation and afforestation[12].

## Discussion

We identified the mix of mitigation strategies that are cost-effective at a given carbon price across regions with developed regions predominantly employing technical options while structural changes through transition toward more intensive but GHG efficient agricultural production systems are projected to be the main source of emission reductions in developing regions. To realize the mitigation potentials presented in this study, several

adoption barriers such as lack of education and infrastructure, poor access to markets or land tenure insecurity[2] will still have to overcome, which will require immediate attention by policy makers. Educating farmers about the positive impacts of, e.g., increasing nitrogen use efficiency or conservation tillage, on individual farmers' welfare, could speed up the adoption of these mitigation practices[26,27]. Even though results imply positive synergies of non-CO$_2$ mitigation for CO$_2$ emissions from land use change, further research is needed to consider explicitly the impact on soil carbon given its importance for the carbon cycle[15]. For example, Sanderman et al.[28] found a strong correlation between soil carbon loss and agricultural land degradation and restoration may enable co-benefits for CO$_2$ sequestration and non-CO$_2$ mitigation in certain areas[15,29].

We find that the adoption of technical mitigation options such as improved rice management (i.e., decreased flooding period of rice paddies) and anaerobic digesters can substantially reduce emissions at low carbon prices as also highlighted by other studies[8,30] and hence should be a priority in agricultural climate change mitigation policies. Widespread adoption of these technical options in developing and developed countries can deliver significant direct non-CO$_2$ emission savings of 0.8 GtCO$_2$eq/year at 100 $/tCO$_2$eq already by 2030. Realizing the mitigation potential from technical options would require modest investment and operation costs of around 12 bn $/year globally by 2030. However, the majority of adoption costs are anticipated to arise in emerging and developing regions. Hence, widespread adoption should be accompanied by technological transfer and support from developed countries. In parallel, efforts need to be pursued to achieve the transition toward more GHG efficient production systems through structural management changes which we identified as a key component for agricultural mitigation efforts. Especially in tropical regions the livestock sector could contribute a very significant amount of non-CO$_2$ abatement (1.75 GtCO$_2$eq/year at 100 $/tCO$_2$eq in 2050) through for example improved feeding and sustainable intensification practices. Herrero et al.[31] showed that increasing energy density of the feed ration in the lest efficient livestock systems by less than 10% allows to reduce GHG intensity by about 50% but also other studies identify a large mitigation potential in the livestock sector[3,9,32]. In addition, results indicate that a positive co-benefit for land use change CO$_2$ mitigation can be expected. In the long run, demand side policies should deliver significant additional

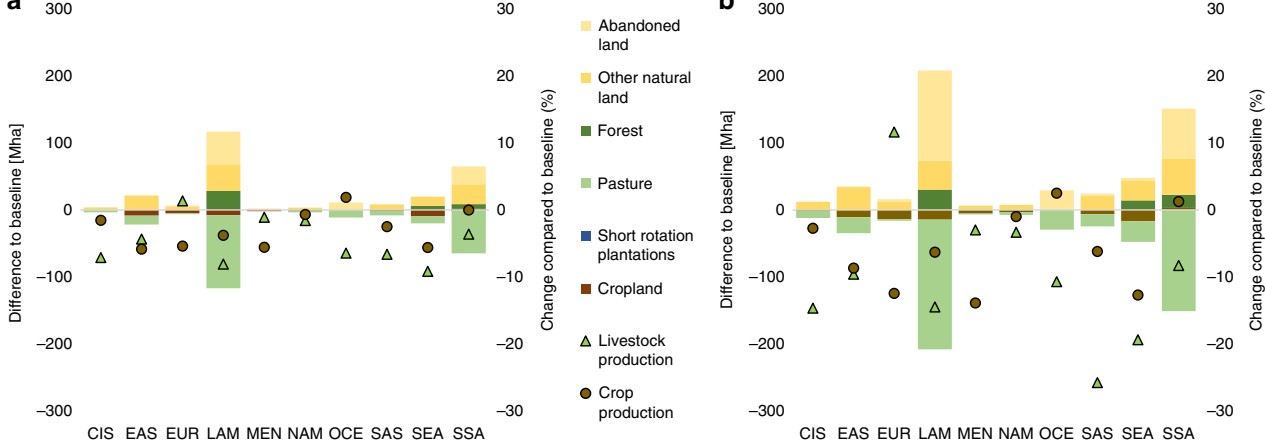

**Fig. 4** Impact of non-CO$_2$ mitigation efforts on land use and agricultural production. Bars present the change in land use in Mha at a carbon price of **a** 40 $/tCO$_2$eq and **b** 100 $/tCO$_2$eq in 2050 compared to the baseline scenario without carbon prices. Single points display the relative change in agricultural production compared to the baseline in 2050. EUR Europe, CIS Commonwealth of Independent States, NAM North America, LAM Latin and Central America, SSA Sub Saharan Africa, MEN Middle East and North Africa, EAS East Asia, SAS South Asia, SEA Southeast Asia, OCE Oceania

mitigation of 0.6 GtCO$_2$eq/year at 100 \$/tCO$_2$eq. Together, the three mitigation wedges would allow achieving an aspirational mitigation target of 1 GtCO$_2$eq/year for agriculture in 2030[14] already at 25 \$/tCO$_2$eq and put agriculture on an emission pathway consistent with a 2 °C target.

## Methods

**Modeling framework.** In this study we apply the GLOBIOM Model[9], a partial equilibrium model of the global agricultural and forestry sectors, including the bioenergy sector. A global agricultural and forest market equilibrium is computed by choosing land use and processing activities to maximize the sum of producer and consumer surplus subject to resource, technological, demand, and policy constraints similar to McCarl and Spreen[33]. Crop- and livestock production is represented with a high spatial resolution at the level of Simulation Units (SimU) going down to 5 × 5 min of arc, and depict different production and management systems, differences in natural resource and climatic conditions as well as differences in cost structures and input use. Here, these units are aggregated to 2 × 2 degrees. The model explicitly covers 18 major crops that together represent over 70% of harvested area and 85% of vegetal calorie supply. Crops are produced in four management systems whose input structure is defined by Leontief production functions parameterized using the bio-physical crop growth model EPIC (Environmental Policy Integrated Model)[34]. The parameterization of the livestock sector is done with the RUMINANT model[31] and initial spatial distribution of ruminants and their allocation between production systems is calibrated to Wint and Robinson[35]. The forestry sector represents the source for logs (for pulp, sawing, and other industrial uses), biomass for energy, and traditional fuel wood, which are supplied from managed forest or short rotation plantations (SRP). Harvesting cost and mean annual increments are informed by the G4M global forestry model[16,36].

Prices are endogenously determined at the regional level to establish market equilibrium to reconcile demand, domestic supply, and international trade. Land and other resources are allocated to the different production and processing activities to maximize a social welfare function which consists of the sum of producer and consumer surplus. The model includes six land cover types: cropland, grassland, short rotation plantations, managed forests, unmanaged forests, and other natural vegetation land. Economic activities are associated with the first four land cover types. Depending on the relative profitability of primary, by-, and final products production activities, the model can switch from one land cover type to another. Land conversion over the simulation period is endogenously determined for each SimU within the available land resources and conversion costs that is taken into account in the producer optimization behavior. Land conversion possibilities are further restricted through biophysical land suitability and production potentials, and through a matrix of potential land cover transitions.

Changes in socio-economic and technological conditions, such as economic growth, population changes, and technological progress, lead to adjustments in the product mix and the use of land and other productive resources. By solving the model recursively dynamic for 10 year time steps, decade-wise detailed trajectories of variables related to supply, demand, prices, and land use are generated. GLOBIOM covers major GHG emissions from agricultural production, forestry, and other land use including CO$_2$ emissions from above- and below-ground biomass changes, N$_2$O from the application of synthetic fertilizer and manure to soils, N$_2$O from manure dropped on pastures, CH$_4$ from rice cultivation, N$_2$O and CH$_4$ from manure management, and CH$_4$ from enteric fermentation. The model explicitly represents technical mitigation options based on Beach et al.[8], structural adjustments in the crop- and livestock sector, i.e., through transition in management systems[9], and consumers' response to market signals[37].

**Technical mitigation options.** We consider the following technical options for crop- and livestock production: optimal fertilization, split fertilization, no-tillage, nitrification inhibitors, residue incorporation, antibiotics, bovine somatotropin, propionate precursors, anti-methanogens, intensive grazing, and various small- to large-scale anaerobic digesters. Rice options are defined as a combination of water- (midseason drainage, continuous flooding, alternative wetting/drying, dry seeding, and dryland rice), residue- (100%/50% residue incorporation and no tillage), and fertilizer management (ammonium sulfate fertilizer, increased/reduced fertilization, optimal fertilization, slow release fertilizer, and nitrification inhibitors).

Percentage emission savings for each technical mitigation option are multiplied by GLOBIOM production system specific (crop- or livestock sector) emission factors to calculate absolute emission savings. We only consider liquid manure from indoor housing to be available for anaerobic digestion. Manure dropped by ruminants directly on pasture are assumed not to be available for digesters. Moreover, we assume that anaerobic digestion is available for all cattle, sheep, and goats including replacement and fattening heifers. Emission savings for propionate precursors were corrected for the time spent on the pasture as this option requires daily administration. For grazing ruminants, we assume one third of the emission saving effect from propionate precursors based on Höglund-Isaksson et al.[38].

Costs for the non-CO$_2$ mitigation options are also based on the mitigation options database from Beach et al.[8]. Costs include annual costs of the different mitigation options (including direct costs and labor costs, change in input costs)

and for certain options, i.e., anaerobic digesters, investment costs. Costs have been extracted in terms of USD per head or ha and do not include revenue changes for farmers due to productivity increase or decrease related to the application of a technology. These revenue changes are endogenously represented in GLOBIOM. Costs for propionate precursors have been corrected for the time animals spend on the pasture. We assume a quadratic cost function where marginal costs double from initial costs at the adoption maximum of a technology. Mitigation options get adopted if the carbon price exceeds the marginal cost of the practice.

With respect to adoption rates of different options we defined mutually exclusive mitigation option bundles for the crop- and livestock sectors. For non-rice crops we assumed that only one option can be applied per ha (full competition between the options). However rice options are defined as a combination of different water, residue, and fertilizer management practices. For the livestock sector we differentiated two bundles: enteric fermentation options and manure management options, where options from both bundles can be implemented at the same time. For Europe, we exclude antibiotics and bovine somatotropin as a mitigation option[39,40] due to current legislation. Since mitigation options are only parameterized for a subset of crops (barley, corn, millet, soybean, sorghum, and wheat; 61% of non-rice cropland) in Beach et al.[8], we assume potential adoption also for other crops represented in GLOBIOM such as beans, chickpeas, potatoes, cassava, cotton, groundnuts, rapeseed, sunflower, sweet potatoes, sugar cane, and oil palm. Supplementary Table 1 and Supplementary Fig. 3 present details on the parameterization of the technical mitigation options and adoption rates in the model.

**Structural mitigation options.** GLOBIOM represents a comprehensive set of management systems at the grid level parameterized using bio-physical models. The EPIC model[34] provides spatially explicit estimates on productivities and input requirements for each of the crop production systems taking into account site specific soil and climate characteristics. Four different crop management systems are differentiated: subsistence farming, low input, high input, and high input and irrigation technology. For the livestock sector, input–output relationships are parameterized using the RUMINANT model[31]. Distinction is made between dairy and other bovines, dairy and other sheep and goats, laying hens and broilers, and pigs. Livestock production activities are differentiated for ruminants into grass based (arid, humid, temperate/highlands), mixed crop-livestock (arid, humid, temperate/highlands), and other systems; for monogastrics into smallholders and industrial systems. For each species, production system and SimU, livestock production is characterized in terms of yields, feed requirements, GHG emissions, manure production and nitrogen excretion. Switches between production systems allow for feedstuff substitution and for intensification or extensification of livestock production. The detailed representation of production systems allows the model to explicitly represent structural changes in the agricultural sector under a climate policy. Farmers can switch to more GHG efficient management practices on site, reallocate production to more productive areas within a region, or through international trade across regions.

**Demand side mitigation.** Demand in GLOBIOM is modeled at the level of 37 aggregate economic regions. Commodity demand is specified as stepwise linearized downward sloped function with constant own-price elasticities following Schneider et al.[41] parameterized using FAOSTAT data on prices and quantities, and own-price elasticities as reported by Muhammad[42]. Final commodity demand is based on the interaction of bioenergy demand, population growth, income per capita growth, and consumer's response to prices. The first three being exogenously introduced in GLOBIOM while consumer's response to prices is computed endogenously. Bioenergy demand projections for agricultural feedstocks are based on Lotze-Campen et al.[43]. Demand for agricultural products increases linearly with population in each region. GDP per capita changes determine demand variation depending on income elasticity values. Income elasticities for agricultural commodities are calibrated to mimic anticipated Food and Agriculture Organization of the United Nations projections of diets[44]. Consumers' response demand to commodity prices is endogenously computed in GLOBIOM based on own-price elasticities which are adjusted to consider GDP per capita growth over time.

Implementing a carbon price in the model may come partly at the cost of food availability if the carbon price acts as a tax on direct emissions from agriculture and impacts agricultural prices and market equilibrium. As a consequence, agricultural commodity prices, especially of GHG-intensive products, i.e., ruminant meat, would significantly increase and consumers' would react by decreasing their consumption levels depending on the price elasticity. As we do not explicitly consider different future diets i.e. reduction of meat consumption, or cross-price elasticities in our analysis, the mitigation potentials related to the consumer's response to price signals may however be overestimated.

**Decomposition of GHG mitigation potential.** We differentiate between three mitigation wedges classified as a bundle of similar mitigation options that are represented in GLOBIOM: technical options (referring to technical options from the EPA[8] database excluding some practices that are already explicitly considered in GLOBIOM, i.e., improved feed conversion efficiency. Intensive grazing is considered as structural options), structural options (which include shift in management systems and reallocation of production through international trade, and intensive grazing), and demand side options (consumer's response to price

changes). Total mitigation was decomposed ex-post to the different wedges applying the formulas presented below. First total mitigation was distributed to the demand side, supply side, and international trade. Mitigation through consumers' response to price changes was calculated by multiplying baseline emissions with average relative emission reductions in the carbon price scenarios compared to the baseline calculated as an average over relative reduction in demand levels and relative reduction in demand levels weighted with the relative change in emission factors.

$$\text{Demand side mitigation}_{r,t,s} = \text{Emissions}_{r,t,s_0}$$

$$* \frac{\left(\left(\frac{\text{CONS}_{r,t,s}}{\text{CONS}_{r,t,s_0}} - 1\right) + \left(\left(\frac{\text{CONS}_{r,t,s}}{\text{CONS}_{r,t,s_0}} - 1\right) * \frac{\text{EF}_{r,t,s}}{\text{EF}_{r,t,s_0}}\right)\right)}{2}$$

Similarly, the mitigation from supply side adjustments (technical and structural) and international trade were calculated (see below). Mitigation from the supply side were further split into direct and indirect emission savings from technical, and structural options by multiplying average relative changes in emission factors (without technical options, including only direct emission savings from technical options, and including emissions savings and productivity changes) with change in production levels and with baseline emissions.

$$\text{Supply side mitigation}_{r,t,s} = \text{Emissions}_{r,t,s_0}$$

$$* \frac{\left(\left(\frac{\text{EF}_{r,t,s}}{\text{EF}_{r,t,s_0}} - 1\right) + \left(\left(\frac{\text{EF}_{r,t,s}}{\text{EF}_{r,t,s_0}} - 1\right) * \frac{\text{PROD}_{r,t,s}}{\text{PROD}_{r,t,s_0}}\right)\right)}{2}$$

$$\text{International trade}_{r,t,s} = \text{Emissions}_{r,t,s_0}$$

$$* \frac{\left(\left(\frac{\text{PROD}_{r,t,s}}{\text{PROD}_{r,t,s_0}} - \frac{\text{CONS}_{r,t,s}}{\text{CONS}_{r,t,s_0}}\right) + \left(\frac{\text{PROD}_{r,t,s}}{\text{PROD}_{r,t,s_0}} - \frac{\text{CONS}_{r,t,s}}{\text{CONS}_{r,t,s_0}}\right) * \frac{\text{EF}_{r,t,s}}{\text{EF}_{r,t,s_0}}\right)}{2}$$

| | |
|---|---|
| CONS | Consumption levels |
| PROD | Production levels |
| EF | Emission factor (non$-CO_2$ emissions/product unit) |
| $r$ | Region (37) |
| $t$ | Year (2000 − 2050) |
| $s_0$ | Baseline scenario without carbon price |
| $s$ | Carbon price scenario 10 − 150 $/tCO_2$eq |

**Scenario analysis**. The baseline scenario represents a business as usual scenario with continuation of current trends with respect to macro- and agronomic drivers until 2050. It is based on the SSP2 from the 5th IPCC Assessment Report[18,45]. World population increases to around 9.2 billion until 2050 and GDP per capita is expected to more than double globally to around 25,000 year-2005 USD per capita. For food demand projections, income elasticities are calibrated to mimic FAO projections of diets[44]. Until 2050, global average calorie intake is projected to increase from around 2900 kcal/cap/day in 2010 to 3300 kcal/cap/day in 2050. Demand for animal protein is relatively high, due to strong income and population growth. Moderate reductions in food waste and losses over time add to the availability of agricultural products. Assumptions on technical change in the crop- and livestock sector follow historic trends[46]. Fertilizer use and costs of agricultural production increase in proportion with yields. Productivity changes through technological change in the livestock sector follow Bouwman et al.[47]. Transition towards more efficient livestock production systems takes place at a moderately fast pace.

To estimate the cost-efficient mitigation potential for agricultural $CH_4$ and $N_2O$ emissions at global level, we apply a carbon price on emissions from agriculture (non-$CO_2$) and land use change ($CO_2$) ranging from 10 to 150 $/tCO_2$eq. The carbon price was implemented 2015 onwards in the objective function as additional cost per $tCO_2$eq emitted and increases linearly over time hitting its maximum in 2050. We estimate the MACC by contrasting results for non-$CO_2$ emissions only from the carbon price scenarios to the baseline without carbon price. In addition, we assess co-benefits with land use change $CO_2$ emissions by implementing a carbon price only on agricultural non-$CO_2$ emissions, and comparing impact on $CO_2$ emissions to the baseline. To limit the decrease in per capita calorie consumption in the carbon price scenarios a food security constraint was implemented. Minimum calorie intake (corresponding to a maximum 1% level of undernourishment) was calculated at the regional level consistent with FAOSTAT reporting of undernourishment based on Hasegawa et al.[48]. This minimum calorie intake acted as a threshold in the carbon price scenarios to avoid negative impacts

on food security under climate mitigation. For regions with lower levels of calorie intake, consumption levels were fixed to the baseline values in the carbon price scenarios. Regions with higher calorie intake were allowed to reduce their calorie intake up to the threshold.

**Data availability**. The authors declare that the main data supporting the findings of this study are available within the article and the supplementary information. Additional data are available upon request from the authors.

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

## Acknowledgements

This work was undertaken as part of the CGIAR Research Program on Climate Change, Agriculture and Food Security (CCAFS). CCAFS is carried out with support from the CGIAR Fund Donors and through bilateral funding agreements. This research has received funding from the European Union's FP7 Project FoodSecure (Grant agreement no. 290693), the European Union's H2020 Project SUSFANS (Grant agreement no. 633692), and CD-LINKS (Grant agreement no. 64214), and technical support from the International Fund for Agricultural Development (IFAD). M.H. acknowledges support from the CSIRO OCE Science Leaders Program and the Belmont Forum/FACCE-JPI DEVIL Project (NE/M021327/1). The views expressed in the document cannot be taken to reflect the official opinions of CGIAR, the US Environmental Protection Agency, or donors.

## Author contributions

S.F., R.B., and P.H. designed the study and performed the research. R.B. processed the input data for technical non-$CO_2$ mitigation options. S.F. did the implementation in GLOBIOM and carried out the model simulations. T.H. and H.V. provided the inputs for the food security constraint. S.F. and P.H. decomposed the mitigation potentials and analyzed together with R.B. the results. S.F. wrote and revised the manuscript with help from R.B., P.H., H.V., M.H., A.M., T.H., J.C., S.R., and M.O.

## Additional information

**Competing interests:** The authors declare no competing interests.

