## [Peer Review File · Nature Communications]

Editorial Note: This manuscript has been previously reviewed at another journal that is not operating a transparent peer review scheme. This document only contains reviewer comments and rebuttal letters for versions considered at Nature Communications. Mentions of prior referee reports have been redacted.

Reviewers' comments:

Reviewer #1 (Remarks to the Author):

The manuscript titled "Mitigation of global non-CO₂ greenhouse gases: How much can agriculture contribute?" estimates a very ambitious mitigation potential of agriculture sector for non-CO₂ GHGs at a lower cost which is very interesting and essential to limit global warming well below 2°C, as agreed in the Paris agreement. Previous estimates i.e. IPCC AR5, Beach et al. (2015)... were lower as compared to present study and this difference was mainly due to inclusion of 2 new measures i.e. structural measures on supply side and consumer's response to price increase on demand side in the current study. The manuscript needs more supplementary/additional information i.e. particularly on methodology section describing the mitigation measures and their adoption potential and also description of probable social and cultural barriers for adoption of included measures. More detailed information on structural supply side mitigation options and contribution of the non CO₂ GHGs and CO₂ sequestration due to land sparing to the total mitigation from structural measures is important to calculate how much is really coming from non-CO₂ GHGs mitigation. This is important because the authors are calculating mitigation potential of non-CO₂ GHG emission and I am not sure if the mitigation arising from land sparing should be included here or not. The manuscript can be considered for publication if supported with supplementary information as mentioned below and the claim that all the emission saving is from non-CO₂ GHG emission is proved. I strongly feel more information on the analysis is needed, even if it is provided as a supplementary material.

Main text (manuscript)

Line 55: The authors projected global agricultural non-CO₂ emissions to increase to around 6.6 GtCO₂eq/year by 2050 under middle of the road SSP2 scenario. It would be good to explain here what diet or per capita consumption the authors considered to estimate this value and how this value compares with emission projections from other studies.

Line 72: The estimation for add-on technologies is 0.8 GtCO₂eq/year at 100\$/tCO₂eq which is higher compared to other studies and author says it is due to better/equal adoption rates across technologies, more detailed information on this and the values of adoption rate would be appreciated.

Line 107: More elaboration on indirect effects of add-on technologies which can contribute about 0.15 GtCO₂eq/year at 100\$/tCO₂eq is required.

Line 111: Structural adjustment contributes 0.9 GtCO₂eq/year at 100\$/tCO₂eq, Havlik et al., 2013 estimated mitigation potential due to autonomous transition towards more efficient systems to be 0.736 GtCO₂eq/year and emission reductions occurred mainly through land sparing but reduction in direct non-CO₂ emission was modest. I am not sure how this compares in the current study and suggest more information should be provided here on the share of mitigation source (non CO₂ and CO₂) when there is a structural change.

Line 115 and 119: What is the difference between technical and technological option? Please define.

Line 147 to Line 159: It is very important to discuss here how the price rise has impacted the consumption i.e. what is the % of price rise and % of change in consumption. Please describe the change in per capita consumption (%) by 2050 and how it is impacted by price rise and also if there is no price rise.

Supplementary material

Line 38: The assumption for crop sector for mutually exclusive mitigation option is "only one option can be applied per ha (full competition between the options)" which to me looks unreal. For example in a rice paddy field emission savings can be achieved by switching to multiple drainage as well as better N management. But from my understanding of above statement authors have calculated the benefits of either of the measures not both. Please clarify.

Reviewer #2 (Remarks to the Author):

This paper adds to growing global evidence on the global mitigation potential from agriculture and uses a global partial equilibrium model to evaluate some of the wider effects potentially arising from implementation of key mitigation across different regions. The model claims to be a more refined partial equilibrium analysis considering more measures than have hitherto been considered in this way; most previous papers having been based on a static analysis of fewer measures considered bottom-up.

The modelling can account for mitigation motivated by supply and demand side measures by i) technical measures ii) structural mitigation options on the supply side, and iii) market feedbacks through consumption and international trade responses to price changes. The second category was most interesting to me (aka "transformational change"). The analysis uses mitigation potential and costs from Beach et al. (2015) for livestock mitigation measures and adds different carbon prices to the objective function to see how GLOBIOM alters the adoption rates of technical measures and how it drives the structural changes, when the model makes the transitions from less to more intensive systems that reduces non-CO2 GHGs.

At first glance there is certainly a new estimate of global economic potential, but a lot is left to the carbon price – as the main policy driver. The paper ends by suggesting that something needs to be done in policy terms (in addition to a carbon price), which the conclusion is reached by other national and international mitigation papers. As such the paper does not move that part of the agenda forward.

As with other mitigation estimations the model assumptions can still be challenged. I observe 3 points to clarify

- Figure 2 International reallocation of production seems a bit unrealistic in many situations. I think it's important to present which share of the "structural" mitigation is due to systems transitions and which proportion is due to reallocation/relocation.

- Page 4 In the remaining regions, technological options account for around one third of the total mitigation potential. While at lower carbon prices mainly improved rice management options and anaerobic (large-scale) digesters are being adopted, intensive grazing (mainly in Latin America). Isn't intensive grazing also represented as a system transition? If so, there is an overlap between the

migration classed as tech. options and that classed as transition. This is difficult for the reader to assess and should be clarified.

- Page 5 Reductions in consumption levels..... In some regions, e.g., in Brazil, this could reduce CH₄ but increase CO₂ emissions if land is not removed from livestock systems – but this is unlikely under the current legislation.

Ultimately this paper leaves us with a more inflated mitigation potential which tries to distinguish the different ways that mitigation will arise including as production is reorganized. This is important for global negotiations. As noted the paper does not take us further forward in term of institutions policy levers and incentives to realize the potential.

Reviewer #3 (Remarks to the Author):

General comments.

This is a well written and potentially interesting analysis, however, I have two major concerns:

1) A justification is required for why this study does not address CO₂ emissions, and what that means for the analysis of technological or structural changes that might increase CO₂ emissions (e.g. soil organic carbon emissions as a result of switching from grazing to crops). This partial analysis seems quite problematic because it only address two of the three major GHGs in agricultural systems.

2) Is that there simply is not enough information about the actual analysis (even including the supplementary materials) for a any degree of replication, or even assurance that the analysis is valid. Effectively a set of results are presented with no really details on how they were produced, the underpinning assumptions, data sources etc. As a reader I had no confidence in how the results presented were arrived at and so, however potentially interesting they may be, I struggle to see how these results are useful. I think much more detail is required regarding the modelling undertaken here. I guess there are restrictions on how much could be put in the main article, but at the very least the SI would need to be significantly expanded to ensure a satisfactory level of transparency, and replicability for this study.

Specific comments

L27 "mitigation wedges" seems like jargon, I would suggest either explaining or substituting this phrase.

L37 in this context the meaning of "5-10 percent reduction" is unclear, a rate of change is converted to an absolute, how?

L44 could it not be true that by ignoring demand side feedbacks could lead to a significant overestimate of the global mitigation potential of agriculture, if the feedbacks are amplifying (e.g. Jevons' paradox/the rebound effect)?

L54 I have not heard of this scenario, why use this rather than a standard IPCC scenario? It means that the findings here are more or less impossible to interpret without understanding what SSP2 is and even then the comparability between this study and others using more ubiquitous scenarios is limited. Some justification and description are needed here.

L67 "allow putting agriculture" awkward phrasing

L71 I am somewhat uneasy that the current study is framed (somewhat implicitly) as a counter to the mainstream consensus on mitigation strategies. That implied consensus is based on a comparison to just three other studies (7, 11, 17). Are these really representative? Are there no meta-studies? Are like for like really being compared given that different scenarios are likely to have been employed across the 4 compared studies. I think these points need to be explicitly addressed in the paper.

L72 "add-on technologies" need to be defined.

L110 what are these structural adjustments that have such a large mitigation potential? Without some details here, I don't find this very informative.

L136-146 My concern here is that the "structural changes" suggested here 1) are not feasible (i.e. soils not suitable etc..) and 2) that the structural changes suggested may reduce CH₄ but will increase CO₂ (particularly related to land use change are soil carbon emissions), which is entirely unaccounted for in this study. Justification for not addressing CO₂ is needed, especially given that later it is claimed that all GHGs are included in GLOBIOM (L197).

L205 do you really mean that "GDP growth per capita" is expected to double, or that GDP is expected to double?

Response to reviewers' comments

We would like to thank all three reviewers for accepting to review this manuscript and the provided valuable and relevant feedback. We have worked hard to address all comments, which we think significantly improved the manuscript, especially with respect to readers' understanding of the model and the transparency on the applied method. For example, we included much more details on the model structure and datasets used, the scenario analysis and drivers, and on the parameterization of mitigation options in the supplementary material and method section in the manuscript. We also included additional detail on potential synergies of non-CO₂ mitigation efforts with CO₂ through avoided land use changes etc. We address each of the reviewer's comments in detail on the pages below.

Reviewer #1:

The manuscript titled “Mitigation of global non-CO₂ greenhouse gases: How much can agriculture contribute?” estimates a very ambitious mitigation potential of agriculture sector for non-CO₂ GHGs at a lower cost which is very interesting and essential to limit global warming well below 2oC, as agreed in the Paris agreement. Previous estimates i.e. IPCC AR5, Beach et al. (2015)... were lower as compared to present study and this difference was mainly due to inclusion of 2 new measures i.e. structural measures on supply side and consumer's response to price increase on demand side in the current study.

The manuscript needs more supplementary/additional information i.e. particularly on methodology section describing the mitigation measures and their adoption potential and also description of probable social and cultural barriers for adoption of included measures.

We added much more information in the supplementary information (SI): i) detailed parameterization of the mitigation options and improved description of implementation in section 3 (costs, impacts on productivities and emissions), ii) a GLOBIOM model description in section 1, iii) extended description of the scenario drivers in section 2, iv) additional results in section 4 e.g. adoption rates of technologies.

In the main text, we further elaborated in the discussion section on barriers for adoption of these measures and give advice on how these could be overcome e.g. improved training of farmers and dissemination as shown by Stuart et al. (2014) and others.

More detailed information on structural supply side mitigation options and contribution of the non CO₂ GHGs and CO₂ sequestration due to land sparing to the total mitigation from

structural measures is important to calculate how much is really coming from non-CO2 GHGs mitigation. This is important because the authors are calculating mitigation potential of non-CO2 GHG emission and I am not sure if the mitigation arising from land sparing should be included here or not.

The agricultural mitigation potentials presented in the paper refer only to non-CO₂ emissions (CH₄ and N₂O) related to crop- and livestock production but do not include, unless stated explicitly, CO₂ effects. Previously, we mentioned only briefly in line 159 the indirect impact of non-CO₂ mitigation on biomass CO₂ changes from land use (+0.7 GtCO₂eq savings in 2050 at 100\$/tCO₂eq under a mitigation scheme targeting only agricultural non-CO₂ emissions). This indirect CO₂ mitigation comes on top of the 2.6 GtCO₂eq/yr from CH₄ and N₂O.

Since the omission of CO₂ impacts and a justification for this was also stressed by reviewer 3, we revised the motivation section in the introduction and explain why we focus on non-CO₂ emissions (they are an important source of emissions and will be in the future one of the key factors, as residual emissions, to determine the absolute level of negative emissions required to achieve ambitious climate stabilization targets). We also included a section in the text where we discuss in more detail co-benefits of non-CO₂ mitigation for land use change biomass CO₂ emissions. We also present land use area changes driven by the GHG price and try to give a first assessment on what impacts on carbon emissions could be anticipated.

The manuscript can be considered for publication if supported with supplementary information as mentioned below and the claim that all the emission saving is from non-CO2 GHG emission is proved. I strongly feel more information on the analysis is needed, even if it is provided as a supplementary material.

We have included additional information on the GLOBIOM model, methods used for this study, and results of our analysis in both the main text as well as the SI. In addition, we have added text to clarify the source of reductions in GHG in terms of non-CO₂ vs. CO₂. As noted above, our primary estimates are for non-CO₂ reductions and CO₂ is included only where mentioned explicitly.

Main text (manuscript)

Line 55: The authors projected global agricultural non-CO2 emissions to increase to around 6.6 GtCO₂eq/year by 2050 under middle of the road SSP2 scenario. It would be good to explain here what diet or per capita consumption the authors considered to estimate this value and how this value compares with emission projections from other studies.

We complemented the methodology section with more information on the methods used to generate agricultural commodity demand projections in GLOBIOM based on the SSP2 data and scenario drivers as well as other data from FAO (i.e. food intake, GDP per capita etc.). In addition, we added text comparing our emissions projections with other studies in more detail. Our estimate on the development of agricultural non-CO₂ emissions is similar to estimates from FAOSTAT for 2050 (around 6.2 GtCO₂eq/year) but slightly more conservative compared to the other integrated assessment models which project emissions to increase to around 7.6 GtCO₂eq/year (IMAGE) to 10.5 GtCO₂eq/year (GCAM).

Line72: The estimation for add-on technologies is 0.8 GtCO₂eq/year at 100\$/tCO₂eq which is higher compared to other studies and author says it is due to better/equal adoption rates across technologies, more detailed information on this and the values of adoption rate would be appreciated.

We added a detailed comparison of the mitigation potential from technologies with existing literature in the main text and explain in more detail the reasons for the difference in the technical mitigation potential compared to Beach et al. (2015). Reasons are indeed the more conservative assumption on adoption shares across technologies but also the limited applicability of mitigation options on a subset of crops in Beach et al. (2015). Looking at relative shares in reduction compared to the Baseline GLOBIOM results are consistent with Beach et al. (2015).

In addition, we added the information on the parameterization of the mitigation options in the SI as requested by the reviewer i.e. costs, GHG impact, productivity impact, and adoption rates.

Line107: More elaboration on indirect effects of add-on technologies which can contribute about 0.15 GtCO₂eq/year at 100\$/tCO₂eq is required.

We added a sentence which explain how (some) options enhance indirectly, via productivity increases, the mitigation potential beyond the direct emission saving.

Line 111: Structural adjustment contributes 0.9 GtCO₂eq/year at 100\$/tCO₂eq, Havlik et al., 2013 estimated mitigation potential due to autonomous transition towards more efficient systems to be 0.736 GtCO₂eq/year and emission reductions occurred mainly through land sparing but reduction in direct non-CO₂ emission was modest. I am not sure how this compares in the current study and suggest more information should be provided here on the share of mitigation source (non CO₂ and CO₂) when there is a structural change.

The mitigation potentials for structural adjustments are consistent in both studies as they rely on the same dataset (Herrero et al., 2013) also used in Havlík et al. (2014). Figure 3 in Havlík et al. (2014), presents a very similar mitigation potential for agricultural non-CO₂ emissions when applying a carbon price of 100 USD/tCO₂eq on the AFOLU sector of around 1 GtCO₂eq from non-CO₂ in 2030.

In general, the mitigation potentials presented throughout the manuscript refer to non-CO₂ emissions only. However, we included an additional section on the positive co-benefits of a non-CO₂ carbon price on mitigation of land use changes and related CO₂ emissions.

Line 115 and 119: What is the difference between technical and technological option? Please define.

This was an imprecision on our side. We use now “technical options” throughout the text and provide a definition in the introduction.

Line 147 to Line 159: It is very important to discuss here how the price rise has impacted the consumption i.e. what is the % of price rise and % of change in consumption. Please describe the change in per capita consumption (%) by 2050 and how it is impacted by price rise and also if there is no price rise.

We added the requested information. Global average calorie intake decreases from around 3,300 kcal/capita/day to around 3,200 kcal/capita/day (-3%) in 2050 at 100 \$/tCO₂eq while average agricultural commodity prices increase by 18%.

Supplementary material

Line 38: The assumption for crop sector for mutually exclusive mitigation option is “only one option can be applied per ha (full competition between the options)” which to me looks unreal. For example in a rice paddy field emission savings can be achieved by switching to multiple drainage as well as better N management. But from my understanding of above statement authors have calculated the benefits of either of the measures not both. Please clarify.

Indeed, this was not properly described in the SI, we clarified this. While for non-rice crops we indeed only allow one i.e. fertilizer option to be applied per ha, for rice, each option is already parameterized in the EPA database as a combination of different water-, residue-, and fertilizer management systems.

Reviewer #2:

This paper adds to growing global evidence on the global mitigation potential from agriculture and uses a global partial equilibrium model to evaluate some of the wider effects potentially arising from implementation of key mitigation across different regions. The model claims to be a more refined partial equilibrium analysis considering more measures than have hitherto been considered in this way; most previous papers having been based on a static analysis of fewer measures considered bottom-up.

The modelling can account for mitigation motivated by supply and demand side measures by i) technical measures ii) structural mitigation options on the supply side, and iii) market feedbacks through consumption and international trade responses to price changes. The second category was most interesting to me (aka “transformational change”). The analysis uses mitigation potential and costs from Beach et al. (2015) for livestock mitigation measures and adds different carbon prices to the objective function to see how GLOBIOM alters the adoption rates of technical measures and how it drives the structural changes, when the model makes the transitions from less to more intensive systems that reduces non-CO2 GHGs.

At first glance there is certainly a new estimate of global economic potential, but a lot is left to the carbon price – as the main policy driver. The paper ends by suggesting that something needs to be done in policy terms (in addition to a carbon price), which the conclusion is reached by other national and international mitigation papers. As such the paper does not move that part of the agenda forward. As with other mitigation estimations the model assumptions can still be challenged.

We significantly revised the conclusion section and tried to put results in a more policy relevant context. We give clearer recommendations which mitigation strategies should be given priority across regions and present in addition cost estimates for the adoption of technical options.

I observe 3 points to clarify

- Figure 2 International reallocation of production seems a bit unrealistic in many situations. I think it's important to present which share of the “structural” mitigation is due to systems transitions and which proportion is due to reallocation/relocation.

We further split the mitigation potential from structural adjustments into transition in management systems and emission savings from international trade/relocation to provide more insights into the results. International trade itself plays only a minor part in the mitigation potential coming from

“structural” adjustments as also reckoned by the reviewer. The majority of the structural mitigation for non-CO2 emissions is indeed related to change in the management and transition of systems.

- Page 4 In the remaining regions, technological options account for around one third of the total mitigation potential. While at lower carbon prices mainly improved rice management options and anaerobic (large-scale) digesters are being adopted, intensive grazing (mainly in Latin America). Isn't intensive grazing also represented as a system transition? If so, there is an overlap between the migration classed as tech. options and that classed as transition. This is difficult for the reader to assess and should be clarified.

We agree with the reviewer. We tried to come up with a better grouping and definition of what we consider as “technical” and “structural” options. In the revised manuscript we report “intensive grazing” as a structural option and provide a definition and explanation on the grouping in the introduction.

- Page 5 Reductions in consumption levels..... In some regions, e.g., in Brazil, this could reduce CH4 but increase CO2 emissions if land is not removed from livestock systems – but this is unlikely under the current legislation.

We are not sure if we understood and hence addressed the comment properly but to give a more comprehensive assessment across CH4, N2O and CO2 we added a paragraph on CO2 implications towards the end of the result section.

Ultimately this paper leaves us with a more inflated mitigation potential which tries to distinguish the different ways that mitigation will arise including as production is reorganized. This is important for global negotiations. As noted the paper does not take us further forward in term of institutions policy levers and incentives to realize the potential.

We updated the conclusion section in order to improve the policy relevance of the paper by giving clear advice on the most favorable options across regions. We think that the detailed regional decomposition of the mitigation potential, which reveals the importance of structural changes in developing and technical options in developed countries, is an important insight for policy makers which provides a strong policy message.

Reviewer #3:

General comments.

This is a well written and potentially interesting analysis, however, I have two major concerns:

1) A justification is required for why this study does not address CO₂ emissions, and what that means for the analysis of technological or structural changes that might increase CO₂ emissions (e.g. soil organic carbon emissions as a result of switching from grazing to crops). This partial analysis seems quite problematic because it only address two of the three major GHGs in agricultural systems.

Indeed, soil carbon emissions are an important source of agricultural GHG emissions which are influenced by the type of management, and local soil and climate conditions. In this paper, we mainly focused on non-CO₂ emissions as the combination of different non-CO₂ mitigation options in a bottom-up modelling framework is the innovative part of our work. Moreover, the dynamic representation of global soil carbon emissions is currently missing in the model (as also in other economic agricultural sector models) and the model covers only above- and belowground biomass changes. However, we agree with the reviewer that CO₂ impacts should at least be discussed so we added some text on this point.

We tried to address the comment by adding information on land use area developments in the mitigation scenarios, which allows to get a first estimate of how soil carbon stocks may change under mitigation policies. Since most pastures decrease due to abandonment and are not converted to arable land e.g. for bioenergy (cropland is also decreasing due to the carbon price incentive), overall soil carbon emissions from agriculture can be expected to decrease due to the revegetation. Moreover, we provide results for biomass CO₂ emissions from land uses change which contribute additional 0.7 GtCO₂/yr in 2050 at 100\$/tCO₂eq and which supports the potential positive synergies with CO₂ mitigation.

We also clarified the GHG coverage in the method section.

2) Is that there simply is not enough information about the actual analysis (even including the supplementary materials) for a any degree of replication, or even assurance that the analysis is valid. Effectively a set of results are presented with no really details on how they were produced, the underpinning assumptions, data sources etc. As a reader I had no confidence in how the results presented were arrived at and so, however potentially interesting they may be, I struggle to see how these results are useful. I think much more detail is required regarding the modelling undertaken here. I guess there are restrictions on how much could be put in the main article,

but at the very least the SI would need to be significantly expanded to ensure a satisfactory level of transparency, and replicability for this study.

We agree with the concern of the reviewers and added more information in the revised SI (see answer to reviewer 1 on the same issue). This includes a detailed model description of GLOBIOM and a list of references to other peer reviewed publications where the model has been applied to similar topic, a description of key modules relevant for the analysis i.e. the parameterization of mitigation options, a more detailed scenario description including drivers and key results, and additional results i.e. baseline emission developments, adoption rates etc. We hope that the additional material enables the reader to build trust in the results presented.

Specific comments

L27 “mitigation wedges” seems like jargon, I would suggest either explaining or substituting this phrase.

We added the definition in the introduction. When speaking about a mitigation wedge in this context we refer to a group/bundle of similar mitigation options. We also are more explicit about the differentiation of technical and structural options in the text.

L37 in this context the meaning of “5-10 percent reduction” is unclear, a rate of change is converted to an absolute, how?

The 5-10% refer to current emission levels. We added this information.

L44 could it not be true that by ignoring demand side feedbacks could lead to a significant overestimate of the global mitigation potential of agriculture, if the feedbacks are amplifying (e.g. Jevons’ paradox/the rebound effect)?

Indeed, this could happen. We reworded the sentence.

L54 I have not heard of this scenario, why use this rather than a standard IPCC scenario? It means that the findings here are more or less impossible to interpret without understanding what SSP2 is and even then the comparability between this study and others using more ubiquitous scenarios is limited. Some justification and description are needed here.

The Shared Socio-Economic Pathway 2 (SSP2) scenario (Fricko et al., 2016) is a “business as usual” scenario of the latest IPCC AR5. It is characterized by continuation of historic trends in e.g. GDP and population growth, and moderate challenges for mitigation and adaptation to climate change. Indeed, for readers not familiar with the IPCC scenario this may create confusion. Hence we replaced SSP2 with “middle of the road” scenario. In brackets we put the reference to SSP2. We provide additional information on the scenario drivers and results in the methodology section and the SI.

L67 “allow putting agriculture” awkward phrasing

Replaced with “setting agriculture”.

L71 I am somewhat uneasy that the current study is framed (somewhat implicitly) as a counter to the mainstream consensus on mitigation strategies. That implied consensus is based on a comparison to just three other studies (7, 11, 17). Are these really representative? Are there no meta-studies? Are like for like really being compared given that different scenarios are likely to have been employed across the 4 compared studies. I think these points need to be explicitly addressed in the paper.

We do not see our study as a counter to the mainstream consensus, but rather an enhancement to incorporate additional sources of mitigation from adjustments in agricultural markets (on both demand and supply sides) in addition to the direct effects of adopting technical options. We substituted the sentence with a more complete comparison with other studies to give a better overview of existing literature. We also elaborate in more detail on the reasons for the higher estimate of technical mitigation options in the current study compared to others which is partly related to the equilibrium modelling approach and integrated representation of different options (Vermont and De Cara (2010) conduct a meta-analysis and find significantly higher non-CO2 mitigation rates in equilibrium models compared to engineering and supply side focused models) but also due to the more narrow focus of some of the studies i.e. Beach et al. (2015) represent only 61% of total cropland.

L72 “add-on technologies” need to be defined.

We replaced add-on technologies with “technical options” throughout the text and provide a definition in the introduction.

L110 what are these structural adjustments that have such a large mitigation potential?

Without some details here, I don't find this very informative.

We added the most important transitions to be more informative.

L136-146 My concern here is that the “structural changes” suggested here 1) are not feasible (i.e. soils not suitable etc..) and 2) that the structural changes suggested may reduce CH4 but will increase CO2 (particularly related to land use change are soil carbon emissions), which is entirely unaccounted for in this study. Justification for not addressing CO2 is needed, especially given that later it is claimed that all GHGs are included in GLOBIOM (L197).

Ad 1) “Structural changes” in GLOBIOM were parameterized using bio-physical models that consider site-specific soil and climate conditions. For example in the crop sector, the EPIC crop model was applied to parameterize for four different management systems (subsistence farming, low input - rainfed, high input – rainfed, high input – irrigated) for 18 different crops. Hence, the suitability of a certain management system for a particular soil type is considered as well as the suitability of a particular site for agricultural production.

Ad 2) We added a justification in the introduction section why we focus on non-CO₂ emissions in the paper and list explicitly in the methodology sections the emission sources covered. We rephrase the sentence where we claimed that “all major emissions are covered”. With respect to CO₂ emissions we only report indirect effects on above- and belowground biomass changes through land use change of the mitigation scenarios but not soil carbon implications. These could not be addressed within this study as they are not represented in the model. Nevertheless, we present carbon price induced area changes and discuss potential implications for soil carbon emissions.

L205 do you really mean that “GDP growth per capita” is expected to double, or that GDP is expected to double?

Indeed we were referring to “GDP per capita” (not GDP growth) which is taken directly from the SSP database (<https://tntcat.iiasa.ac.at/SspDb/dsd?Action=htmlpage&page=about>). We corrected the wording. In the scenarios (SSP2) world population is projected to grow from 6.9 billion people in 2010 to 9.2 billion people in 2050. GDP is projected to increase from around 67 to 230 trillion USD2005 (PPP) by 2050 resulting in a GDP per capita increase from around 10 to 25 thousand USD2005 on global average.

References

- Beach, R. H., J. Creason, S. B. Ohrel, S. Ragnauth, S. Ogle, C. Li, P. Ingraham and W. Salas (2015). "Global mitigation potential and costs of reducing agricultural non-CO₂ greenhouse gas emissions through 2030." Journal of Integrative Environmental Sciences **12**(sup1): 87-105.
- Fricko, O., P. Havlik, J. Rogelj, Z. Klimont, M. Gusti, N. Johnson, P. Kolp, M. Strubegger, H. Valin, M. Amann, T. Ermolieva, N. Forsell, M. Herrero, C. Heyes, G. Kindermann, V. Krey, D. L. McCollum, M. Obersteiner, S. Pachauri, S. Rao, E. Schmid, W. Schoepp and K. Riahi (2016). "The marker quantification of the Shared Socioeconomic Pathway 2: A middle-of-the-road scenario for the 21st century." Global Environmental Change **42**: 251-267.
- Havlík, P., H. Valin, M. Herrero, M. Obersteiner, E. Schmid, M. C. Rufino, A. Mosnier, P. K. Thornton, H. Böttcher, R. T. Conant, S. Frank, S. Fritz, S. Fuss, F. Kraxner and A. Notenbaert (2014). "Climate change mitigation through livestock system transitions." Proceedings of the National Academy of Sciences **111**(10): 3709-3714.
- Herrero, M., P. Havlík, H. Valin, A. Notenbaert, M. C. Rufino, P. K. Thornton, M. Blümmel, F. Weiss, D. Grace and M. Obersteiner (2013). "Biomass use, production, feed efficiencies, and greenhouse gas emissions from global livestock systems." Proceedings of the National Academy of Sciences **110**(52): 20888-20893.
- Stuart, D., R. L. Schewe and M. McDermott (2014). "Reducing nitrogen fertilizer application as a climate change mitigation strategy: Understanding farmer decision-making and potential barriers to change in the US." Land Use Policy **36**: 210-218.
- Vermont, B. and S. De Cara (2010). "How costly is mitigation of non-CO₂ greenhouse gas emissions from agriculture?" Ecological Economics **69**(7): 1373-1386.

Reviewers' Comments:

Reviewer #1:

Remarks to the Author:

I have gone through the additional information supplied, the authors have taken care of all the points raised. They have also clarified few points of concern clearly. I am happy to recommend the manuscript to be accepted for publication.

Reviewer #2:

None

Reviewer #3:

Remarks to the Author:

In general a good job has been done in addressing the points raised in my review. For me just two minor points remain:

1) Some of the edits have led to awkward phrasings, unnecessary repetition, undefined acronyms etc. (e.g. L24, L44, L83, L96, L152, L211-212, L242) so some final proof reading is still required.

2) I still believe that a short, explicit discussion on the limitations of a model on GHG abatement that does not include all CO₂ sources would significantly strengthen the paper. This would only need be a couple of sentences. Without such a discussion the results are open for misinterpretation.

The first point is relatively trivial and the second was not an issues raised by the other reviewers so I would leave it to the editors to make a final decision on the necessity of these change. For my perspective not addressing these issues would not be a bar to acceptance for publication. I do not need to see a further iteration of manuscript.

REVIEWERS' COMMENTS:

Reviewer #1 (Remarks to the Author):

I have gone through the additional information supplied, the authors have taken care of all the points raised. They have also clarified few points of concern clearly. I am happy to recommend the manuscript to be accepted for publication.

Reviewer #3 (Remarks to the Author):

In general a good job has been done in addressing the points raised in my review. For me just two minor points remain:

1) Some of the edits have led to awkward phrasings, unnecessary repetition, undefined acronyms etc. (e.g. L24, L44, L83, L96, L152, L211-212, L242) so some final proof reading is still required.

We added the missing definitions and revised the wording.

2) I still believe that a short, explicit discussion on the limitations of a model on GHG abatement that does not include all CO₂ sources would significantly strengthen the paper. This would only need be a couple of sentences. Without such a discussion the results are open for misinterpretation.

We included a brief discussion in the conclusion section.

The first point is relatively trivial and the second was not an issues raised by the other reviewers so I would leave it to the editors to make a final decision on the necessity of these change. For my perspective not addressing these issues would not be a bar to acceptance for publication. I do not need to see a further iteration of manuscript.